# Powdery Mildew of Bigleaf Hydrangea: Biology, Control, and Breeding Strategies for Resistance

**Christina Jennings** [1], **Fulya Baysal-Gurel** [1] and **Lisa W. Alexander** [2,*]

1   College of Agriculture, Tennessee State University, Nashville, TN 37209, USA
2   United States Department of Agriculture-Agricultural Research Service, U.S. National Arboretum, Floral and Nursery Plants Research Unit, McMinnville, TN 37110, USA
*   Correspondence: lisa.alexander@usda.gov

**Abstract:** *Hydrangea macrophylla*, commonly known as bigleaf, garden, French, or florist hydrangea, is the most economically important member of the *Hydrangea* genus, with a breeding history spanning hundreds of years. Bigleaf hydrangea breeding improvement has largely focused on aesthetic traits and there are few varieties tolerant or resistant to major diseases such as powdery mildew. Powdery mildew is an obligate biotrophic Ascomycete in the order *Erysiphales* represented by approximately 900 species worldwide. The disease-causing agent in hydrangeas is *Golovinomyces orontii* (formerly *Erysiphe polygoni* DC), which tarnishes the beauty, growth, and salability of bigleaf hydrangea plants, especially those packed closely in production environments. Chemical or biological control is commonly used in production. A recently published haplotype-resolved genome of bigleaf hydrangea enables targeted analyses and breeding techniques for powdery mildew resistance. Analyzing transcriptomes of tolerant and susceptible hydrangeas through RNA sequencing will lead to the identification of differentially expressed genes and/or pathways. Concurrent application of marker-assisted selection, genetic transformation, and gene editing will contribute to the development of powdery-mildew-resistant varieties of bigleaf hydrangea. The aim of this review is to give a general overview of powdery mildew, its impact on bigleaf hydrangea, current control methods, molecular mechanisms, and breeding prospects for powdery mildew resistance in bigleaf hydrangea.

**Keywords:** disease resistance; *Golovinomyces orontii*; *Hydrangea macrophylla*; mlo genes





## 1. Introduction

*Hydrangea* species are historically important plants used worldwide for ornamental horticulture [1–6]. *Hydrangea* is a genus of flowering shrubs with approximately 80 species, the most popular being *Hydrangea macrophylla* [3,4,6]. *Hydrangea macrophylla*, commonly known as bigleaf, garden, French, or florist hydrangea, is the most economically important member of the *Hydrangea* genus, accounting for over USD 155 M total sales in the United States in 2019 [7]. The breeding history of bigleaf hydrangea spans hundreds of years, with aesthetic improvements being the primary focus [5]. They are commonly used as in-ground plants, pot plants, and as cut flowers in the floriculture industry. Bigleaf hydrangea thrive within USDA Hardiness Zones 6 to 9 [8]. These plants are native to China, Japan, and East Asia, can grow three to seven feet in height, and are known for their large, colorful inflorescences [9,10]. Plants typically bloom for two to six weeks in late spring, with some varieties blooming continuously into autumn. The inflorescences of bigleaf hydrangea are either mophead or lacecap. Mopheads have a spherical inflorescence, while lacecap inflorescences are flat and round [9]. The inflorescence of mophead cultivars have numerous sterile flowers on the outside and few fertile flowers in the interior of the inflorescence. Lacecap cultivars have an outer ring of a few, showy, sterile flowers with an inner ring of many fertile flowers [11–13]. There are over 700 cultivars, with about 150 of those available in the United States trade [14]. Disease tarnishes the beauty, growth, and

salability of many ornamental plants, including bigleaf hydrangea [15–17]. Plant pathogens that negatively impact ornamental crops are bacteria, fungi, and viruses. Approximately 70% of plant diseases are caused by fungi, which often leads to a decrease in yield and resultant economic loss. Many plant fungal pathogens can be classified into the phyla Ascomycota and Basidiomycota [18]. One common disease affecting bigleaf hydrangea that falls into the phyla Ascomycota is powdery mildew [5,8]. Powdery mildew symptoms on bigleaf hydrangea are more severe in tightly packed production areas or highly shaded areas. Disease resistance for bigleaf hydrangea is desired, either by traditional breeding or engineering resistance via molecular means [14]. The aim of this review is to give a general overview of powdery mildew, its impact on bigleaf hydrangea, current control methods, molecular aspects of powdery mildew resistance, and breeding prospects for resistance in bigleaf hydrangea.

## 2. Powdery Mildew

Powdery mildew is a frequently encountered widespread disease that affects many different mono- and dicotyledonous plants and is one of the most important diseases of many food crops and ornamentals [19–22]. There are presently 16 to 80 genera with approximately 900 species worldwide that are known to cause powdery mildew disease [20,23]. All species are obligate biotrophs of vascular plants, which are comprised mostly of dicotyledons [23]. Powdery mildew consists of different species, with each species having a limited host range [24,25]. Powdery mildew is an ascomycete in the order Erysiphales [14]. These fungi spread readily, adapt through a short life cycle, and have the possibility of sexual recombination [26].

Powdery mildew is easily recognizable [15]. The fungal infection appears first as faint circular white spots that spread into mats that have the potential to cover most plant organs. Powdery mildews have been found in a wide range of environments, which include arid, subarctic, temperate, and tropical habitats [27]. These fungi favor warm days and cool nights, with temperatures of 15° to 25 °C being preferred, respectively [28]. Many powdery mildews involved with ornamental crops thrive in shady conditions with high relative humidity above 75% [29]. However, high humidity appears to not be favorable for dispersal of conidia [19,30]. The life cycle of a particular powdery mildew species will typically be aligned with the host plants [23].

There are many unknowns about powdery mildews. Due to their biotrophic obligate nature, powdery mildews are not able to be cultured, which hinders research efforts. Recent research indicates that the diversity of powdery mildews and their biology are much more complex than previously realized, with the complete life cycles of most species being unknown [19,31]. Information on the ultrastructure of conidiogenesis is also lacking [19]. A commonly studied powdery mildew is *Blumeria graminis* (DC.) Speer (Figure 1), which infects grasses. However, *B. graminis* differs from other powdery mildews, such as the powdery mildew that infects *H. macrophylla*, which could cause limitations in knowledge overall about these fungi. Despite this, most of what is known about powdery mildews is based on *B. graminis* [19]. Other economically important powdery mildews that appear on ornamental crops are *Podosphaera pannosa* of rose [18], *Erysiphe pulchra* of dogwood [32], and *Erysiphe australiana* of crape myrtle [33].

Powdery mildews are unique in that they develop mostly epiphytically [23]. These fungi are pleomorphic, which means that they form multiple morphologically distinctive spore states. They can be asexual (anamorph) and/or sexual (teleomorph), with both life cycles spreading by conidium or ascospore [15,19], respectively, that land on a plant surface and develop structures that penetrate the host cell wall (Figure 2). Spore germination and infection observed in *B. graminis* happened within 60 s of the conidium landing on the host [19]. Spores have been observed being fastened to the cell by liquid extracellular material with cutinase and esterase activity. Immunolabeled antigen from conidia were observed in 30–90 min within the host cell wall [19]. This occurs by the conidium germinating on the adaxial surface of a plant and a short primary germ tube being produced, typically

within 30–60 min, that produces a cuticular peg that penetrates the cuticle of the plant. The primary germ tube then induces the production of the appressorium, which forms about 10 h after infection [19]. The appressorium is a specialized structure that penetrates the cuticle and cell wall of the plant epidermal cells via narrow protrusions. Protrusions by appressoria are responsible for penetrating the host plasma membrane. As a result of enzymatic activity and turgor pressure, a hyphal peg penetrates the host epidermal cell and forms the haustorium [19,20]. Powdery mildew has dedicated infection structures that are called haustoria, which are specialized to their biotrophic nature. Haustoria are enlarged extensions of pegs, which develop inside colonized host cells and absorb nutrients from the host [19,34,35]. Haustorium also play a role in establishing and maintaining a relationship with the infected host plant [15]. Mycelium and conidiophores, which form chains of conidia, cause an unappealing fuzzy gray growth on the plant surface that is visible to the naked eye [34]. Conidia's capability of germinating in a water-free environment may be due to being single monokaryotic cells that have water-filled vacuoles [23]. Infection can spread via air, plant-to-plant contact, and water splash.

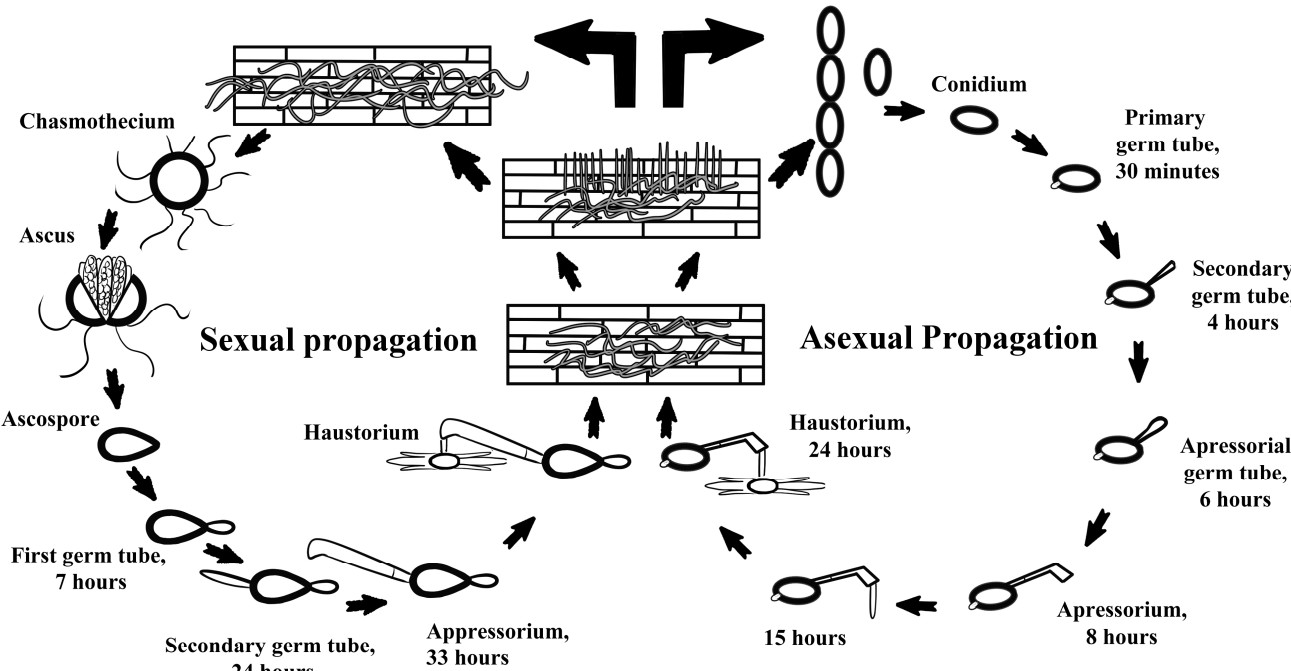

**Figure 1.** Illustration of the *Blumeria graminis* infection process showing sexual propagation (**left side of image**) and the asexual propagation process (**right side of image**).

Perennation is the method in which the pathogen persists through unfavorable conditions [18]. Powdery mildews have three primary methods of perennation. Chasmothecia, the sexual structures, can survive cold winters and hot, dry summers, which are not conducive to powdery mildew growth [36]. Once conditions are favorable, the chasmothecia release ascospores that will infect susceptible plants [15]. The fungus can also overwinter in dormant plant buds [29,37]. The buds can contain hyphae with haustoria, condiophores, and conidia so, once dormancy breaks, "flag shoots" can continue the disease cycle. Mycelia can also persist even in unfavorable conditions, which is another type of perennation [19]. In bigleaf hydrangea, powdery mildew generally overwinters as spores of fungal hyphae and disease in greenhouses able to be active all year without the need for perennation [38].

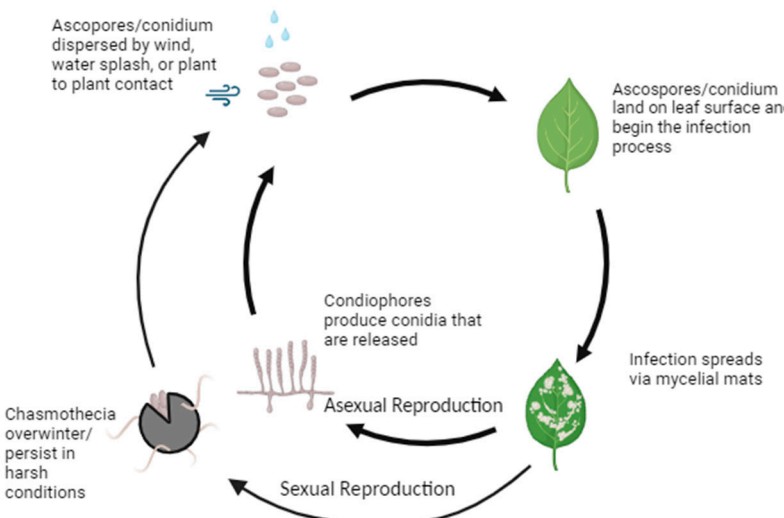

**Figure 2.** Basic scheme of *G. orontii* life cycle demonstrating the common asexual reproduction stage and the rarer sexual reproduction stage involving chasmothecia. Created with Biorender.com (accessed on 21 December 2023).

## 3. Powdery Mildew in Hydrangea

The powdery mildew disease-causing agent in bigleaf hydrangea is *Golovinomyces orontii* (formerly *Erysiphe polygoni* DC) [38–40]. The name change occurred because there were extensive studies in the taxonomy of powdery mildew fungi [16]. Powdery mildew taxonomy and identification were largely based on teleomorph characteristics. It was later found that the anamorphic forms are taxonomically important. Molecular phylogenetic studies provided information which divided into five major lineages. Analyses revealed the polyphyletic nature of *Erysiphe* and new genera were introduced, including *Golovinomyces*, which was previously *Erysiphe* sect. *Golovinomyces* [41]. *Golovinomyces orontii* has a wide host range and worldwide distribution. In the first report of *G. orontii* in hydrangea, fungal conidia germinate and form poorly developed to nipple-shaped appressoria. Fang et al. observed *Euoidium*-type germination and no chasmothecia in that specific observation [19,42,43]. This specific powdery mildew typically occurs as the asexual morph and rarely as the sexual morph [43]. *Golovinomyces orontii* is an ectoparasite that produces conidiophores epiphytically and vegetative mycelium [15]. The mycelium of *G. orontii* can occur on both sides of leaves, as well as on stems. The hyphae are typically straight to sinuous. The condiophores of *G. orontii* are solitary and arise from the hyphal mother cells or towards the end of the cell [39]. Around 2 h after inoculation, the conidium develops its primary germ tube. Four hours after inoculation, the primary appressorium forms at the tip of the primary germ tube. Twelve hours after inoculation, secondary germ tubes are then started from the conidium and primary appressorium with septa that separate the conidium and primary appressorium. A day after inoculation, there is growth of secondary germ tubes, which present as hyphae with septa and lateral appressoria. Three days after inoculation, multiple hyphae emerge from a conidium, forming a branched hyphal structure that will subsequently develop into a lesion [44]. Severe infection can cause plant growth to slow or stop entirely (Figure 3) [25,38]. While powdery mildew is not typically considered fatal to bigleaf hydrangea, infection can lead to extensive chlorosis or yellowing of the leaves, premature defoliation, and leaf area and shoot elongation reduction [14]. Disease symptoms tend to be more severe on plants that are in shaded areas with high humidity and limited air movement, such as greenhouses [8,37,38]. *Golovinomyces orontii* can persist throughout the year in favorable conditions and has the ability to overwinter in the form of fungal hyphae or spores attached to the plant or in plant debris [25,38]. This fungus is

economically important due to powdery mildew's ability to detrimentally affect hydrangea production that often takes place in tightly packed areas [14,34]. Plants that are tolerant or resistant to powdery mildew are more desirable to reduce fungicide use, protect worker health, reduce funds spent on protecting bigleaf hydrangea health, and decrease the risk of fungicide-resistant pathogens [5,14,20,45].

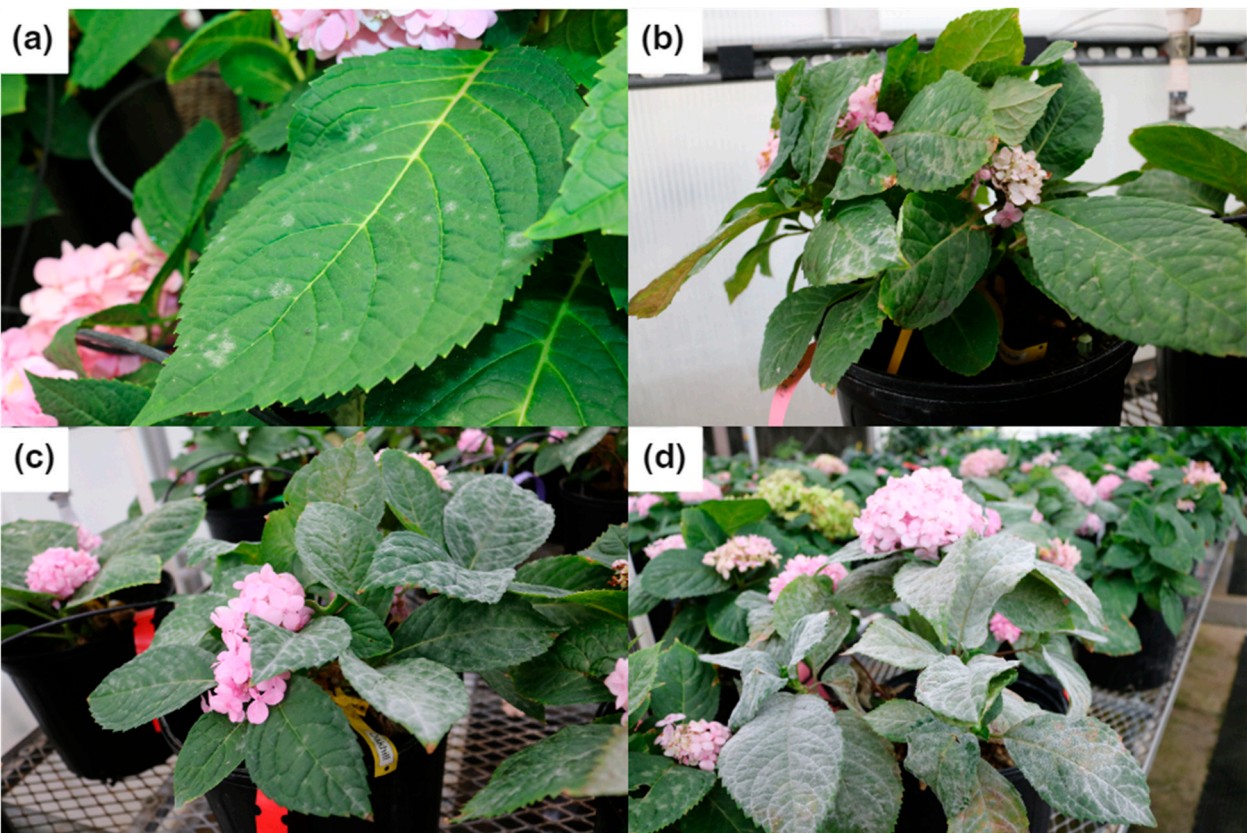

**Figure 3.** Development of *G. orontii* on *Hydrangea macrophylla*. (**a**) Low infection, (**b**) moderate infection, (**c**) moderate to high infection, and (**d**) severe powdery mildew infection.

## 4. Control of Powdery Mildew

Bigleaf hydrangea cultivars range in susceptibility to powdery mildew infection [39,46]. With this knowledge of *H. macrophylla*, several methods can be incorporated to manage powdery mildew in production settings, which include biological, chemical, and cultural.

### 4.1. Biological Control

Biological control methods stand out for their effectiveness in minimizing adverse environmental impacts [47]. They have been employed under field and greenhouse conditions using fungal and bacterial antagonists [48]. Biorational products can be adopted for fungal disease management. These products refer to pesticides of natural origin such as botanicals, minerals, microorganisms, and minimum-risk chemicals that have reduced or no negative effects on the environment or beneficial organisms [49,50]. Biocontrol of powdery mildew was achieved on various hosts, including *Trichoderma* isolates [23,51], yeasts, mycophagous arthropods, mycolytic bacteria, and additional effective biological agents. *Ampelomyces quisqualis* Ces [23] is a well-known antagonist species of powdery mildew. *Bradyrhizobium japonicum* symbiotic signal molecules were reported to reduce the size of powdery mildew spots as well as infection incidence [23]. A mycophagous mite, *Orthotydeus lambi*, was found to be able to suppress powdery mildew of grapevines [52]. Plant extracts are another option to induce host resistance. These plant extracts can cause an increase in enzymes that play a defense role against pathogens [48,53]. It is also suggested that root symbiosis with

rhizobia can be used by priming plants with salicylic acid accumulation and defense-gene expression, which is triggered by powdery mildew [23]. However, the downside to certain biological control and biorational products is the limitation of efficacy in wider field conditions [50]. There is also the concern of non-native species posing an ecological risk [47]. These methods open up new avenues of research that can be applied and studied on *G. orontii* of bigleaf hydrangea.

### 4.2. Chemical Control

The most common form of chemical control of powdery mildew used in production is by use of fungicides. These pesticides specialize in chemicals that target fungi and their spores by killing and/or preventing their growth [24,40]. Fungicides can be used in different production environments such as shade houses, greenhouses, or row crops. Fungicides with different modes of action (MOAs) can be incorporated into a fungicide rotation program, which can vary in the treatment regimen dependent on disease severity [40]. The mode of action refers to how the active ingredient(s) in a pesticide impede the target pest [54]. In fungicides, the MOAs typically work by reducing/stopping spore production, germination, and growth of the pest by blocking a specific metabolic pathway [55]. Choosing an appropriate fungicide is crucial. Certain fungicides may target a specific pathogen, while others can be used as a broad range. Various studies have shown successful control of powdery mildew on bigleaf hydrangea using fungicides [8,40,56]. Sulfur, neem oil, triforine, and potassium bicarbonate are commonly used for chemical control of powdery mildew [57]. However, there are drawbacks to the use of fungicides as chemical control. Just because a fungicide can be used to control a specific pathogen does not mean it can be used on every host species. If the fungicide used is not appropriate for the treated plant, phytotoxicity can occur. Phytotoxicity can result in a nonmarketable plant or death of plants [58]. The cost of repeating fungicide applications can be a limiting factor in the ability to control fungal infection in this manner. For realistic and cost-effective applications, it is essential to monitor the disease actively rather than relying solely on preventative measures [23]. Fungicides applied preventatively provide the best control of powdery mildew symptoms by stopping the infection from developing. However, complete prevention is still a challenging goal. Downsides to applying fungicides preventatively include the economic impact to growers and the environmental impact from more fungicide usage. If symptoms appear, the best control will be achieved by fungicide usage as soon as possible [25]. Timing for these types of applications must be precise or powdery mildew incidence could continue to increase. Additionally, many fungicides have also been prohibited in the European Union, making them less accessible for use. There are also many concerns, such as environmental impacts and disease-resistant strains from improper fungicide use. Nonetheless, certain nonfungicidal products, such as chitosan, have been employed. These products possess promising commercial value as they offer broad-spectrum plant protection in an environmentally friendly manner [23]. Chemical control methods can also benefit from being used in combination with cultural control methods [57].

### 4.3. Cultural Control

Cultural control methods optimize plant health by using good horticultural practices, such as proper cultivation, fertilization, irrigation, and sanitation. Providing ideal conditions for optimal plant growth and development will reduce plant stress, which decreases the chance of disease incidence [25]. Proper cultivation means that the plants are having their specific conditions for healthy growth met. If these conditions are not met, the likelihood of pest issues increases. Similarly, fertilization and irrigation need to be tailored to the specific species. Too little fertilization can result in nutrient deficiencies and too much can result in excessive growth of new shoots, which could be more susceptible to plant pathogens [24,37,50]. The host plant growth rate increases with the level of nitrogen available. Specifically, it stimulates fresh, tender growth in plants, which can more easily

be infected. Therefore, nitrogen fertilizer should be avoided during the late summer period to help avoid infection [25,59]. While overhead irrigation may be good to help reduce or slow down the spread of powdery mildew [24], overwatering plants can result in diseases and the lack of oxygen in overwatered conditions may cause root cells to collapse, causing the plant to become more susceptible to disease, such as powdery mildew. However, too much overhead irrigation may raise the relative humidity, causing more ideal conditions for fungal development [25]. Sanitation methods aid in limiting the amount of pathogen inoculum and decreasing fungal diseases. Removing dead plant material and routine pruning of old leaf tissue can discard possible future sources of inoculum [60,61]. This material should be eradicated to ensure the effective termination of the fungus. Pruning infected or overcrowded plant material can help to remove the disease and increase air circulation [57]. Disinfecting and sterilizing surfaces and equipment, such as pruners, is key to limiting potential spread of inoculum [50]. However, cultural control methods require both knowledge of crop and pest biology and how they interact. They typically take long-term planning for the greatest impact, which requires the growers to know when to implement these practices. These methods may additionally require more labor depending on whether multiple methods are needed with this type of control and how many crops must be maintained. Time and resources must also be invested in training any potential new workers [62]. Another form of control is that of host resistance. Bigleaf hydrangea cultivars have varying tolerance and susceptibility to powdery mildew [14,46,63].

## 5. Molecular Mechanisms of Powdery Mildew Resistance

Many studies have found that qualitative resistance is less durable and robust than quantitative resistance against pathogen evolution. The genetic status of both pathogens and host plants plays a role in the outcome of the interaction [26]. Plants are able to protect themselves from pathogens through genetic resistance. Some plants, such as Arabidopsis and barley, have resistance to powdery mildew through loss-of-function mutant alleles of mildew resistance locus O (MLO). This is also known as an impaired S gene. The MLO genes are conserved within the plant kingdom [64]. The proteins reside within the plasma membrane and have one C-terminal calmodulin-binding domain and seven transmembrane domains. Wild-type plants that lack the MLO proteins display both resistance to powdery mildew fungal infection and dysregulated cell death control [65]. Mildew resistance locus O-based resistance is effective against a vast majority of powdery mildew isolates and, in previous studies, has proven to be durable [45,66]. Wild types could be a potential source of more genetic diversity to incorporate into research goals, since the genetic diversity within cultivated bigleaf hydrangea is low [67]. This indicates that, in future research of powdery mildew of bigleaf hydrangea, genes that are prevalent within the plant kingdom can be incorporated.

Pathogen-associated molecular patterns (PAMPs) and damage-associated molecular patterns (DAMPs) are the initial defense by the plant immune system after pathogen attack [68]. Two major types of plant immune responses include host and non-host resistance to pathogens. Host resistance tends to be less durable and is accomplished by a single or multiple resistance I genes. Selection pressure in these scenarios often leads to pressure on pathogens to adapt and evade detection by the host [69,70]. Another example of resistance to powdery mildew includes introgression. For example, five resistance genes from wild tomato species were introgressed into a susceptible tomato cultivar, creating near-isogenic lines (NILs). Two of these genes, Ol-4 and Ol-6, had a unicellular hypersensitive response (HR), which led to complete resistance to powdery mildew. Three of the genes, Ol-1, Ol-3, and Ol-5, had incomplete resistance [71]. Plants also have the ability to impede fungus development by modifying their transporter systems to move sugars away from infected cells. Orthologs of the Sugar Transport Protein 13 (*STP13*) subfamily orthologs appear to play a role in this modification [23]. Once the fungal infection is established, there is a hypersensitive response in the form of programmed cell death, which provides a defense

mechanism in numerous species and we can use this information to help us successfully identify underlying genes responsible for a resistant/susceptible response in hydrangea.

Selecting cultivars that exhibit resistance to the specific pathogen, if available, helps mitigate the spread of pathogens to susceptible plants [58]. Pinpointing the genetic differences between a susceptible and tolerant genotype can help to understand how a disease works within a plant and to eventually treat the disease or aid in developing resistant plants [72]. RNA sequencing (RNA-Seq) is an analysis technique based on next-generation sequencing (NGS) that provides measurement, identification, and comparison of gene expression in the target transcriptome [73,74]. This technology provides a way to profile all of the expressed genes of an individual at a given time point by sequencing complementary DNA (cDNA) translated from the complete messenger RNA (mRNA) of a given tissue or cell type. This method can examine the quantity and sequences of RNA as well as which genes are turned on or off [75]. RNA-Seq also provides accurate results at a low cost. Expression levels measured by RNA-Seq are represented by discrete counts and are highly reproducible [76]. When there is a statistical difference in read counts or expression levels between two conditions for a gene, that is when the gene can be called a differentially expressed gene [74]. Differentially expressed genes (DEGs) provide a way to pinpoint candidate biomarkers back to the genome [72].

DEGs conferring powdery mildew resistance are known/being studied in several other species, including *Arabidopsis* [77,78], rose [79], apple [80], tomato [81], pea, pepper, tobacco, and bread wheat [64]. *Arabidopsis* utilizes the AtMLO2 phylogenetic clade, which is a co-ortholog of Mlo in barley [82]. Another functional ortholog for powdery mildew resistance is the PMR4 gene in *Arabidopsis* and the SIPMR4 gene in tomato. Both genes allow resistance to the tomato powdery mildew, *Oidium neolycopersici* [81]. Using previous studies, such as those with Arabidopsis, corresponding resistance genes can be identified in other species and applied to current research [83].

*RNA-Seq to Find DEGs in Hydrangea for Future Biotech Use*

Transcriptomes are complete sets of transcripts that can be found in cells or tissues for a specific developmental stage or physiological condition. Analyzing the transcriptomes can aid in identifying DEGs so that new genes and/or pathways can be more fully understood [68,84]. RNA-seq technology allows for transcriptomes to be analyzed so that large-scale gene expression datasets can be created for future research [85]. Using different cultivars of *H. macrophylla* that have varying levels of tolerance or susceptibility, tissue samples can be taken at different time points after powdery mildew inoculation in order to compare up- and down-regulated genes using RNA-seq. Once these cultivars are compared at different time points, up-regulated responses to infection can be identified. Knowing what molecular events are occurring will aid in marker-assisted selection to develop disease-resistant varieties [68].

## 6. Prospects for Breeding for Powdery Mildew Resistance in Hydrangea

### 6.1. Evaluating PM Resistance of Cultivars, Species, and Hybrids

Accurate, reproducible phenotyping is the first step toward targeted breeding of powdery mildew resistance in *Hydrangea*. Powdery mildew fungi synchronize their reproductive and developmental phases with the life cycle of the host plant, so understanding how the parasite/host interaction operates in their unique environments is crucial [19]. Hypersensitive response in the form of programmed cell death has been described in numerous species and has been largely used in breeding resistance to powdery mildew [23]. Within the genus Hydrangea, there is a diversity of responses to powdery mildew infection. An example of this is *Hydrangea macrophylla* being more susceptible to infection than *Hydrangea quercifolia* or *Hydrangea febrifuga* (previously *Dichroa febrifuga*) [38,86].

Disease susceptibility quantification is necessary to characterize plant genes that contribute disease resistance or susceptibility. This presents a challenge in powdery mildews due to the overall unculturable nature of the fungus. There have been many reported

inoculation methods of powdery mildews, which include brushing spores directly onto the desired host plant, spraying a spore suspension, using a vacuum-operated settling tower, and spore delivery by the combination of a nylon mesh membrane and sound-based vibrations [21]. Detached leaves can be used to grow powdery mildews in vitro and maintained in a lab environment [19]. Another method to inoculate is to create single-spore spots developed by using an eyelash glued to a wooden stick [87]. However, many of these methods all come with disadvantages. Using the spore-brushing method is an easy method with inconsistent doses of powdery mildew inoculum delivery. Using a spore suspension provides more consistent and even coverage but can result in poor spore germination due to reduced viability of spores after the water-suspension process. The vacuum-operated settling tower and spore delivery through a nylon mesh membrane method are able to achieve even inoculation; however, they are not flexible in the number of plants that can be inoculated in a single event and are restricted to lab use [88]. There is also a mesh-based inoculation box spore-brushing method, which is flexible to a degree and able to be used outside of a lab setting. However, this can be impeded by material cost and total size of plants combined with a large number of plants [21]. For the leaf-based bioassay, leaves must be maintained in good conditions or powdery mildew transfers will have to take place more often [87].

*6.2. Breeding for Powdery Mildew Resistance*

Once a phenotyping system is in place, conventional and genome-enabled breeding strategies can be used to improve PM resistance in bigleaf hydrangea. Conventional breeding strategies most commonly used in bigleaf hydrangea breeding include recurrent selection, backcrossing, and wide hybridization [5,11,89–95]. Recurrent selection uses controlled crosses between a susceptible and resistant parent to produce a large F1 population that is evaluated for PM resistance. The top 1–5% of plants are retained. These F1 selections are then intercrossed (recurrent selection) or crossed to the resistant parent (backcrossing). In these strategies, the mean number of favorable alleles increases in each generation, and plants may be selected based on the trait only or using an index that combines the trait and other ornamental characteristics [96]. Selection usually happens in the F1 generation because hydrangeas are not self-fertile and will not readily produce production-quality F2 plants. Many off-target progenies in the F1 generation mean that large populations in the thousands are necessary to make progress [93,97]. For PM breeding, progress is limited because there is no genetic resistance to PM within the species; the best F1 progeny are moderately tolerant [46]. From pollination to final selection, the process of F1 and BC1 selection take approximately 5 and 8 years, respectively (Figure 4A).

Combining different species to create hybrids, such as crossing *H. macrophylla* and *H. febrifuga*, can provide a way to obtain novel plants with desired characteristics such as powdery mildew resistance [46,92]. This can be observed through evaluation trials where plants are observed for disease severity over a set amount of time. In a greenhouse evaluation, *H. febrifuga* × *H. macrophylla* F1 and BC1 hybrids had less powdery mildew development compared to *H. macrophylla* cultivars over multiple years with varying degrees of powdery mildew pressure. Interspecific hybridization provides an opportunity to understand which crosses and hybrids provide the best powdery mildew resistance and to develop new, resistant cultivars [46]. Genetic resistance, after being established, is the cheapest and most efficient strategy to combat disease. Resistant plants are also a more environmentally friendly option [23].

Marker-assisted and genome-enabled breeding can reduce the time to new variety development and the number of progenies to evaluate in woody ornamental breeding programs [93,95,97,98]. Tools for marker-assisted selection and genome-enabled breeding strategies in bigleaf hydrangea include a full-length, annotated genome for cultivars Endless Summer and Veitchii [99], a high-density genetic linkage map [97], and genetic markers for inflorescence shape [98,100] and double flowering [99,101]. High-quality SSR and SNP markers have been developed and used in diversity analysis and genetic mapping [98,100,102–105]. Tran-

scriptomic analyses have been used to identify genes involved in aluminum tolerance and accumulation [106–108], flower development [109], leaf color [110,111], and stress response [112]. Current studies on transcriptional changes following inoculation with powdery mildew will provide a rich resource to target candidate genes and candidate pathways for introgression into superior cultivars.

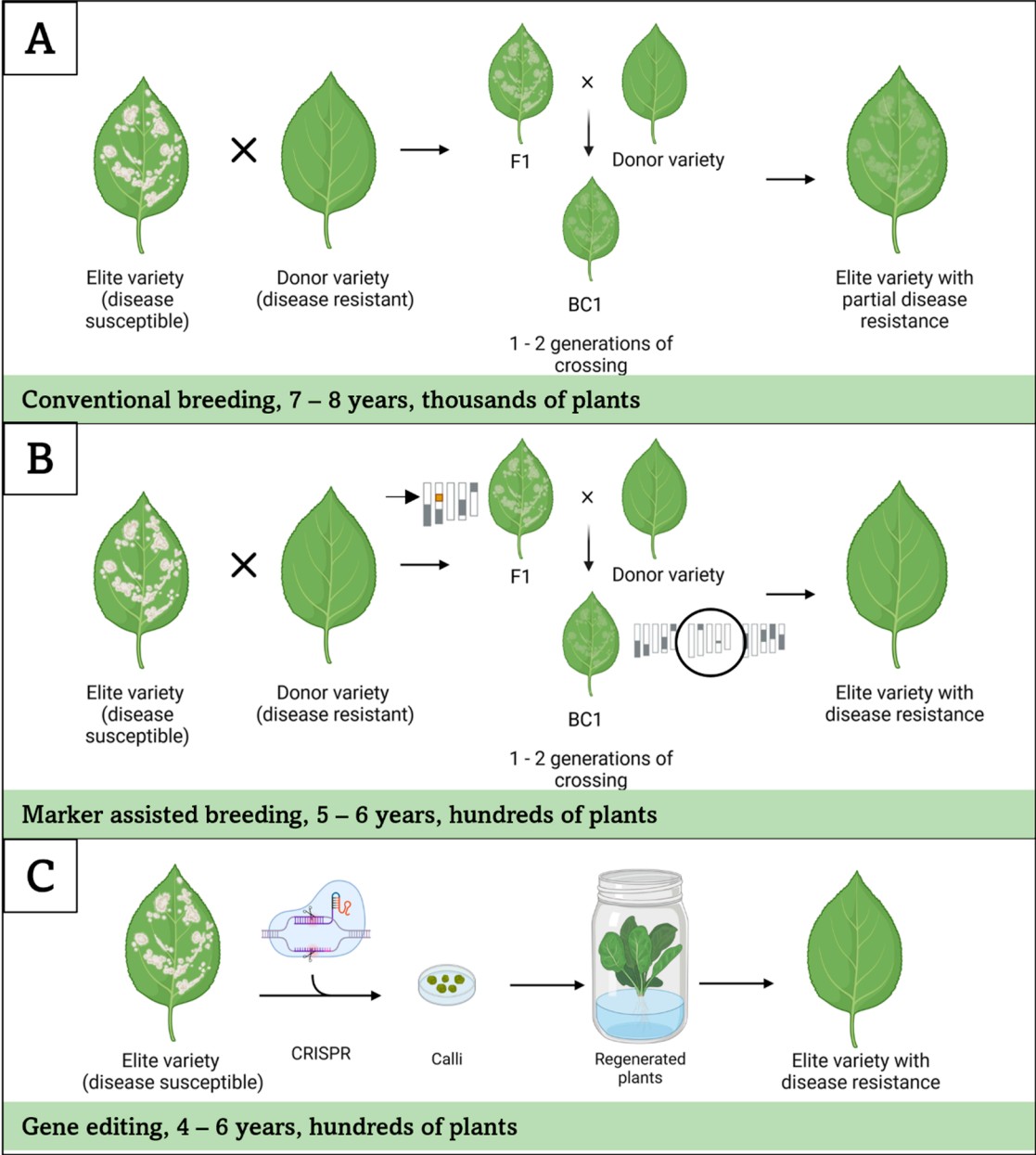

**Figure 4.** Comparison of strategies for developing bigleaf hydrangea varieties with resistance to powdery mildew. (**A**) Conventional breeding with selection at the F1 or BC1 generations; (**B**) marker-assisted breeding with foreground selection in the F1 generation and background selection in the BC1 generation; (**C**) precision breeding using CRISPR gene editing. Created with Biorender.com (accessed on 8 January 2024).

Innovative combinations of foreground selection, background selection, and gene pyramiding may be used to introduce genetic PM resistance into production-quality bigleaf hydrangea cultivars. Foreground selection targets a gene or QTL related to the trait of interest. It is useful for cases where the trait is difficult to evaluate or recessively inher-

ited [113]. Background selection uses unlinked markers scattered throughout the genome to select against the donor parent and maximize the recovery of the recurrent parent genome. Combining these methods in young F1 seedlings would lead to the selection of desirable genes or QTLs, such that alleles can be fixed in a homozygous state and allow selection against undesirable allele combinations and the donor parent genome (Figure 4B). In this way, hundreds (all containing desirable trait alleles) rather than thousands of progenies can be evaluated for PM resistance in each generation. Gene pyramiding strategies use controlled crosses among several parents, each containing a unique allele, to develop a population fixed for two or more alleles. Gene pyramiding is often used for combining multiple disease resistance genes for specific races of a pathogen [18,114,115]. The number of genes strongly influencing powdery mildew resistance and the diversity of those genes within cultivated bigleaf hydrangea will determine which modern breeding strategies can most effectively be used to produce a bigleaf hydrangea cultivar with durable powdery mildew resistance.

*6.3. Engineering Powdery Mildew Resistance in Bigleaf Hydrangea*

Genome editing methods are desirable in ornamental species, since they often have high heterozygosity, high chromosome number, large genomes, long life cycles, and can be polyploid [116,117]. Transgenic technology allows for genes to be transferred to a host plant from any source and produce plants more quickly than traditional breeding techniques. There are currently three major transformation techniques, which are *Agrobacterium*-mediated, biolistic, and protoplast transformation. *Agrobacterium*-mediated transformation is currently the first choice in the development of transgenic plants due to being the most robust and easiest method in the case of many plants. This transformation technique uses *Rhizobium tumefaciens* bacterium to code arbitrary transgenes using transfer DNA (T-DNA). Gene editing technology can precisely modify target genes. In this system, genetically engineered nucleases generate site-specific breaks in the genome and induce the organism to repair the breaks through natural DNA repair methods [118–121]. Gene editing improves flower breeding efficiency, shortens breeding cycles, and reduces the number of plants to evaluate compared to traditional breeding (Figure 4C). To date, most CRISPR/Cas9 use in ornamental plants has focused on flowering and floral regulation, such as increasing insensitivity to ethylene in rose, reducing ethylene synthesis, and prolonging the flowering period of *Chrysanthemum* [121,122]. CRISPR gene editing technology has been used to improve disease resistance in Arabidopsis and tobacco, but quantitative traits in woody ornamental plants with complex genomes have not yet been applied. A major hindrance to the adoption of gene-editing in hydrangea is the lack of an efficient regeneration system. The time and difficulty of calli induction and regeneration poses a hurdle in any species [122] but regeneration of bigleaf hydrangea is especially difficult due to low rates of callus induction and poor shoot formation [123–126]. Tissue-culture-free delivery systems that both solve the drawback of traditional transformation methods and reduce off-target effects are being explored [127,128].

**7. Conclusions**

Powdery mildew still has mysteries despite being a highly researched plant disease. The economic importance, not only to bigleaf hydrangea but to multiple food and ornamental crops, means that further research is needed to elucidate this disease. More research of *G. orontii* would be greatly beneficial to further understand bigleaf hydrangea tolerance and susceptibility. While there are many control methods, such as chemical, biological, and cultural, more resistance options among bigleaf hydrangea cultivars would be a boon. Conventional breeding is a timely task and takes up large amounts of space for plants and materials. Molecular work of *G. orontii* on bigleaf hydrangea would greatly benefit researchers and provide further insight into this plant disease. Additionally, little RNA-seq research of *Hydrangea macrophylla* affected by powdery mildew has been conducted, leaving more information to be desired. New genomic tools for bigleaf hydrangea will enable

modern breeding strategies that can effectively be used to produce a bigleaf hydrangea cultivar with durable powdery mildew resistance.

**Author Contributions:** Conceptualization, L.W.A. and F.B.-G.; methodology, L.W.A., F.B.-G. and C.J.; writing—original draft preparation, C.J.; writing—review and editing, C.J., L.W.A. and F.B.-G.; visualization, C.J. and L.W.A.; supervision, L.W.A. and F.B.-G. All authors have read and agreed to the published version of the manuscript.

**Funding:** This work was supported by the Floral and Nursery Research Institute (FNRI) and the US Department of Agriculture, Agricultural Research Service (USDA-ARS) in-house project #8020-21000-086-000D. Mention of a trademark, proprietary product, or vendor does not constitute a guarantee or warranty of the product by USDA and does not imply its approval to the exclusion of other products or vendors that also may be suitable.

**Data Availability Statement:** Not applicable.

**Conflicts of Interest:** The authors declare no conflicts of interest.

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
