# Peer review of "Powdery Mildew of Bigleaf Hydrangea: Biology, Control, and Breeding Strategies for Resistance"

_horticulturae, doi:10.3390/horticulturae10030216_

Round 1

Reviewer 1 Report

Comments and Suggestions for Authors

Compliments on a very good review paper.

There are only a few minor issues to be solved.

The host species and the pathogen were sufficiently elaborated in the introduction section as well as subsequent chapters.

At the end of the introduction section please add the review papers aims. 

Cultural Control: please mention the drawbacks, such as too much labor needed, which can be reluctant. 

Wild relatives could be more elaborated as a source of resistance/tolerance genes.

Line 307: was the biolistic method applied in hydrangea? Or any similar flowering crop?

Author Response

At the end of the introduction section please add the review papers aims. 

Thank you, information has been added into the introduction to state the aim of the review paper. (L53-57)

Cultural Control: please mention the drawbacks, such as too much labor needed, which can be reluctant. 

Thank you for the comment, information has been added to the cultural control section to mention drawbacks to these methods. (L234-240)

Wild relatives could be more elaborated as a source of resistance/tolerance genes.

Thank you for the comment, a section has been added into section 5 to describe the benefits of wild types for resistant/tolerant genes.  (L255-257).

Line 307: was the biolistic method applied in hydrangea? Or any similar flowering crop?

Thank you for your comment. We could not find and example of the biolistic method in a woody nursery crop. We edited this section to reduce the methods of transformation and focus on the applications of, and barriers to, transgenic breeding and gene editing in hydrangea (lines 418 - 442).

Reviewer 2 Report

Comments and Suggestions for Authors

This study is aimed to review powdery mildew of bigleaf hydrangea including biology, control, and strategies for breeding improvement. The review concept is acceptable. The review can be a useful contribuition for Readers. However, some aspects need suitable revisions before possible conideration for publications.

Suggestions:

L44: At the end of the Introduction, you should give a well defined objectives of your review.

L45-99: In this section, you try to give a general overview of powdery mildews. You may give a large emphasise on Bumeria graminis (which infect wheats – not a horticultural crop). Maybe it is better to chose a powdery mildew species from a horticultural crops.

Fig. 1. Your general mildew life cycle can be missleading as many powdery mildews behave differently. Maybe, it would be more useful if you give the life cycle of your powdery mildew fungus: Golovinomyces orontii.

L124: The control section gives quite limited information. The structure is quite limited (biological, chemical and cultural). In addition, especially the cultural control section provides many general statements with only 1 citation. The stucture can be improved such as non-chemical control measures: agronomic measures, mechanical, physical and biological control, cultivar resistance. Agronomic measures: cropping system and cover crop, plant material and planting, pruning and phytotechnics, soil management, and nutrient supply. Physical control methods: pruning, removal of inoculum sources, shredding, burying etc.

L318: Conclusion and future prospects.

Please povide some future propect issues that were drawn form your study.

Author Response

Reviewer 2

L44: At the end of the Introduction, you should give a well defined objectives of your review.

Thank you for the comment, objectives have been clarified at the end of the introduction. (L53-57 )

L45-99: In this section, you try to give a general overview of powdery mildews. You may give a large emphasise on Bumeria graminis (which infect wheats – not a horticultural crop). Maybe it is better to chose a powdery mildew species from a horticultural crops.

Thank you for your comment. Bumeria graminis is emphasized due to its heavily studied nature. A statement has been made addressing commonly studied powdery mildew species of ornamentals as well. (L:86-88)

Fig. 1. Your general mildew life cycle can be missleading as many powdery mildews behave differently. Maybe, it would be more useful if you give the life cycle of your powdery mildew fungus: Golovinomyces orontii.

The figure has been updated from saying ”Basic scheme of the powdery mildew life cycle.“ To ”Basic scheme of G. orontii life cycle.”

L124: The control section gives quite limited information. The structure is quite limited (biological, chemical and cultural). In addition, especially the cultural control section provides many general statements with only 1 citation. The stucture can be improved such as non-chemical control measures: agronomic measures, mechanical, physical and biological control, cultivar resistance. Agronomic measures: cropping system and cover crop, plant material and planting, pruning and phytotechnics, soil management, and nutrient supply. Physical control methods: pruning, removal of inoculum sources, shredding, burying etc.

Thank you for the thorough review, we added 9 citations to the cultural control section. We have improved the structure so that relevant control measures are discussed, Measures such as irrigation, pruning, sanitation, fertilizers, removing inoculum sources, and resistant cultivars are discussed. (L213)

L318: Conclusion and future prospects.

Thank you for your comment, conclusion was expanded on and future prospects were listed (L444).

Please povide some future propect issues that were drawn form your study.

We greatly expanded section 6, which was renamed “Prospects for breeding powdery mildew resistance in hydrangea” to emphasize the future prospects for breeding that our current capabilities are enabling (lines 318 - 442). We greatly expanded the conclusions section to tie together each section of the paper in the framework of resistance breeding (lines 445 – 457). 

Reviewer 3 Report

Comments and Suggestions for Authors

General comments

The purpose of the review article is to summarize the data from studies of powdery mildew on ornamental plant species bigleaf hydrangea. The content is divided into six sections, which are presented in a proper order. However, the information included in some parts is general and/or does not match directly to the title of the section. This makes the review insufficiently clear and detailed. To make the overview more comprehensive and focused on the topic, additional data and examples relevant to the subject are needed. It is necessary to reorganize the content of individual parts and include more specific information related to powdery mildew and hydrangea.

The overview is based on literature data from 61 sources. Due to the fact that the article is a review paper, the included references are insufficient. The powdery mildew disease has been investigated in many aspects by a huge number of researchers and an extremely large amount of data has been accumulated. Expanding the review with additional data according to the suggestions and recommendations will lead to a qualitative improvement of the revised version. The main accent of the paper should be addressed to application and achievements of innovative breeding techniques to increase the resistance of hydrangea against powdery mildew to be suitable for the topic of Special issue.

Specific comments

In the Abstract there is a contradiction between two statements in:

“Bigleaf hydrangea breeding improvement has largely focused on aesthetic traits and there are few varieties tolerant or resistant to major diseases such as powdery mildew” (rows 10-11) and “Chemical or biological control is commonly used in production as there are no cultivars of bigleaf hydrangea that are resistant to powdery mildew” (rows 15-16). So, it is not clear there is or there is not any resistance to powdery mildew.

The Introduction is too short and does not give a full picture of the state of the problem and the need of such overview. It should be expanded to cover all aspects of the manuscript.

The biology of powdery mildew disease in the section 2 in described in detail.

The section 3 is sort, but informative and well illustrated disease symptoms on hydrangea by pictures of Figure 2.

The title of section 4 (Control of powdery mildew in Hydrangea production) is very specific, but the data and examples are more or less common, not related to the topic of this part and in general. Therefore the information should be expanded to correspond to the title of the section or the caption has to be changed to only “Control of powdery mildew”. All examples for Biological control are related to other plant species (pea, zinnia, vines; Ref. 15, 26, 28), not to bigleaf hydrangea or other disease on hydrangea as botrytis blight (Ref. 27). Studies on Biological control of powdery mildew on hydrangea are missing. The section for Chemical control mainly contains basic information about fungicides and the disadvantages of their application. The information is not directed to examples relating to powdery mildew control. There is one paper for Chemical control of powdery mildew of bigleaf hydrangea that is in press (Ref. 29). Similar comment for Cultural control – general information is presented and only one study for the control of Botrytis cinerea in ornamental production is citied (Ref. 33). Addition information focused on control of powdery mildew should be included in this part of the manuscript.

The section 5 is informative and examples in are related to the topic.

Into section 6, there are mixed general data with some specific information that have no direct link to breeding for powdery mildew resistance in hydrangea. For instance, there is too detailed information about different methods for the inoculation methods of powdery mildews, the RNA-seq technology of rose is mentioned, but real examples for breeding of hydrangea are missing. It is not clear, what is the connection between content under title “Regeneration/transformation systems available” and resistance of hydrangea to powdery mildew. This part should include at least information about the achievements and experience regarding regeneration, transformation and genome editing techniques applied to hydrangea, not just general information. The title of this section probably should be not “Breeding for powdery mildew resistance in hydrangea”, but “Prospects for breeding….”.

The conclusions are not exhaustive and do not correspond to the content of manuscript. The main statement in the section is that “Little RNA-seq research of Hydrangea macrophylla effected by powdery mildew has been conducted”, but this does not stand out as the most significant achievement from the presented data.

References are up to date, but are insufficient to prepare a detailed overview, especially for such important plant disease that is intensively and thoroughly investigated.

Author Response

 Reviewer 3

The purpose of the review article is to summarize the data from studies of powdery mildew on ornamental plant species bigleaf hydrangea. The content is divided into six sections, which are presented in a proper order. However, the information included in some parts is general and/or does not match directly to the title of the section. This makes the review insufficiently clear and detailed. To make the overview more comprehensive and focused on the topic, additional data and examples relevant to the subject are needed. It is necessary to reorganize the content of individual parts and include more specific information related to powdery mildew and hydrangea.

The overview is based on literature data from 61 sources. Due to the fact that the article is a review paper, the included references are insufficient.

Thank you for this important comment, we added 63 new references including 40 addressing genome-enabled breeding strategies and tools,

The powdery mildew disease has been investigated in many aspects by a huge number of researchers and an extremely large amount of data has been accumulated. Expanding the review with additional data according to the suggestions and recommendations will lead to a qualitative improvement of the revised version. The main accent of the paper should be addressed to application and achievements of innovative breeding techniques to increase the resistance of hydrangea against powdery mildew to be suitable for the topic of Special issue.

Thank you for your comment. We greatly expanded Section 6. “Prospects for breeding powdery mildew resistance in hydrangea” by adding several new topics including comparisons of conventional vs. genome-enabled breeding for PM resistance, tools available for genome-enabled breeding of hydrangea, and application of markers in innovative selection strategies (lines 319-442), including over 40 new literature citations. We added a new figure, Figure 4, the provides a visual comparison of three breeding strategies for PM resistance in hydrangea (line 880). These sections bring together previous ideas presented in the manuscript and interpret them in light of current capabilities and future prospects for breeding hydrangea PM resistance.  

Specific comments

In the Abstract there is a contradiction between two statements in:

“Bigleaf hydrangea breeding improvement has largely focused on aesthetic traits and there are few varieties tolerant or resistant to major diseases such as powdery mildew” (rows 10-11) and “Chemical or biological control is commonly used in production as there are no cultivars of bigleaf hydrangea that are resistant to powdery mildew” (rows 15-16). So, it is not clear there is or there is not any resistance to powdery mildew.

Thank you for the comment. Statement “as there are no cultivars of bigleaf hydrangea that are resistant to powdery mildew” was removed for clarity.

The Introduction is too short and does not give a full picture of the state of the problem and the need of such overview. It should be expanded to cover all aspects of the manuscript.

Thank you for the comment, the Introduction has been expanded and objectives have been clarified at the end of the introduction. (L53-57)

The title of section 4 (Control of powdery mildew in Hydrangea production) is very specific, but the data and examples are more or less common, not related to the topic of this part and in general. Therefore the information should be expanded to correspond to the title of the section or the caption has to be changed to only “Control of powdery mildew”.

Thank you for your comment. The title of section 4 has been updated to only say “Control of powdery mildew.”

All examples for Biological control are related to other plant species (pea, zinnia, vines; Ref. 15, 26, 28), not to bigleaf hydrangea or other disease on hydrangea as botrytis blight (Ref. 27). Studies on Biological control of powdery mildew on hydrangea are missing.

Thank you for the comment, unfortunately the literature appears to be lacking in specific biocontrol of powdery mildew of hydrangea. We are using studies on other organisms that can then be related back to hydrangea.

The section for Chemical control mainly contains basic information about fungicides and the disadvantages of their application. The information is not directed to examples relating to powdery mildew control. There is one paper for Chemical control of powdery mildew of bigleaf hydrangea that is in press (Ref. 29). Similar comment for Cultural control – general information is presented and only one study for the control of Botrytis cinerea in ornamental production is citied (Ref. 33). Addition information focused on control of powdery mildew should be included in this part of the manuscript.

Thank you for your comment. We have expanded both the cultural and chemical control sections to include more references and information. There are now 11 references in the chemical control section and 10 references in the cultural control section. (L182,212)

Into section 6, there are mixed general data with some specific information that have no direct link to breeding for powdery mildew resistance in hydrangea. For instance, there is too detailed information about different methods for the inoculation methods of powdery mildews,

We greatly expanded section 6, removed some details regarding PM inoculation, and rearranged the text to focus on hydrangea breeding for PM resistance (lines 319-442). Methods of PM inoculation are now part of a larger section on breeding, where they are presented as a necessary first step as part of an accurate and reproducible phenotyping system.

The RNA-seq technology of rose is mentioned, but real examples for breeding of hydrangea are missing.

This section was removed and placed as a “future prospect” in section 5. “Molecular mechanisms of powdery mildew resistance (lines 305-316). Candidate genes, linked markers, and other outputs from transcriptional studies are listed in lines 390-394, but as yet have not been used in a completed breeding improvement project.

It is not clear, what is the connection between content under title “Regeneration/transformation systems available” and resistance of hydrangea to powdery mildew. This part should include at least information about the achievements and experience regarding regeneration, transformation and genome editing techniques applied to hydrangea, not just general information.

This section was greatly expanded to include applications of, and barriers to, transgenic breeding and gene editing in hydrangea, as suggested (lines 418-442).

The title of this section probably should be not “Breeding for powdery mildew resistance in hydrangea”, but “Prospects for breeding….”.

Changed as suggested

The conclusions are not exhaustive and do not correspond to the content of manuscript. The main statement in the section is that “Little RNA-seq research of Hydrangea macrophylla effected by powdery mildew has been conducted”, but this does not stand out as the most significant achievement from the presented data.

Thank you. We greatly expanded the conclusions section to tie together each section of the paper in the framework of resistance breeding (lines 445 – 457). 

References are up to date, but are insufficient to prepare a detailed overview, especially for such important plant disease that is intensively and thoroughly investigated.

Thank you for this important comment, We added 63 new references including 40 addressing genome-enabled breeding strategies and tools.

Reviewer 4 Report

Comments and Suggestions for Authors

The title of the manuscript is’ Powdery Mildew of Bigleaf Hydrangea: Biology, Control, and Breeding Strategies for breeding improvement; however, 90 % of the manuscript content seems to focus more on general information about powdery mildew on various hosts rather than providing in-depth discussions on the specific host and pathogen in the context of hydrangea. I recommend addressing this discrepancy to ensure alignment between the title and the actual content of the manuscript. My specific comments can be found in the attached MS.

Comments on the Quality of English Language

My comments can be found in the attached MS. 

Author Response

Reviewer 4

The title lacks specificity regarding the aspect of breeding improvement. It can be changed to “Powdery Mildew of Bigleaf Hydrangea: Biology, Control, and Breeding Strategies for Resistance.”

Title corrected as recommended, thank you.

I would replace this with obligate biotroph (L50)

Thank you for your comment, ‘parasites’ has been changed to ‘obligate biotroph.’ (L63 )

Please use the metric system (L59)

Thank you for the comment, corrected.

Reference please!! (L59)

Reference added as suggested, thank you.

Not in all cases, though; for example, Podsphaera cerasi conidial production decreased drastically at high relative humidity (RH) (L60)

Thank you for your comment. Moparthi et al: “Podosphaera cerasi - an old foe of US sweet cherry with a new name – its biology, epidemiology, and beyond” stated that 45% to 85% was optimal but over 85% was not. Due to this statement, we believe ‘above 75%’ still holds true.

I am particularly concerned about the aspects that are currently unknown. Could you provide more details or elaborate on the specific areas of powdery mildews where our understanding is lacking, and further research is needed (L63)

Thank you for the comment, the statement has been expanded on to incorporate unknowns, including the inability to culture powdery mildew as it is an obligate biotroph which hinders research efforts, and that the biology of powdery mildew is much more complex than originally thought (L79-82)

Reference please. (L64)

Thank you for the comment, reference specified.

Reference please (L72)

Thank you for the comment, reference specified.

Is it host cell wall? If so, please mention that. (L74)

Thank you for your comment, the statement has been clarified by adding ‘host’.

Reference please (L78)

Thank you for your comment, reference has been clarified.

This is a vague statement, could you please provide more context or details? (L81-82)

Thank you for the comment, statement was expanded on to describe haustoria and to add clarity to the original statement that it is a specialized structure for biotrophs. (L103-107)

Please correct the spelling here. (L94)

Thank you for the comment and catching the misspelling, spelling has been corrected.

Please add reference here. (L96)

Thank you for the comment, references expanded on and added.

Reference, please. It would be helpful to provide examples of different powdery mildew species that overwinter using various methods of perennation. (L96)

Thank you for the comment, more references added as well as the topic of perennation expanded on to further describe survival in harsh conditions(L118-126).

I do not find any new information in this section apart from the one above. Either merge these two sections or provide more details on the biology of G. orontii. (L100)

Thank you for the comment, more information has been added about G. orontii. (L133-139)

Instead of the isolates displaying – I would replace with – conidia germinate and form nipple shaped appressoria. (L102-103)

Thank you for the comment, the text has been corrected to reflect the suggestion.

It would be valuable for the authors to include information on condidial germination and the formation of hyphal appressoria by the fungal isolates. (L 102)

Thank you for the comment, additional information of conidial germination and hyphal appressoria has been added. (L131-137)

Repetition, already mentioned this above, unless you start the sentence as G. orontii…(L 109-111)

Thank you for the comment, the sentence was changed to start with Golovinomyces orontii.

Citation please (L109)

Thank you for the comment, a citation has been added.

The statement is repetitive; please merge it with the one above in line 111 to avoid redundancy (L 112-113)

Thank you for the comment, statement has been deleted and previous statement expanded to merge and to avoid redundancy.

You discussed this in the sections below. Are there any specific details on the biology of G. orontii that you want to address here (L113-114)

Thank you for the comment and pointing out the repetition, the sentence was deleted to avoid repetition.

Do you want to display different stages of G. orontii infection on H. macrophylla (I would change the title accordingly) or is it intended to demonstrate various cultivar responses to G. orontii infection? (Figure 2; L120)

Thank you for the comment, it is not intended to represent different cultivar responses. ‘Powdery mildew’ was replaced by ‘G. orontii' for the sake of clarity.

Severe (Figure 2; L122)

Thank you for the comment, ‘displays nearly complete foliar coverage by..’ replaced by ‘severe.’

Did you mean cultivars?(L125)

Thank you for the correction, yes, we meant cultivars instead of species. The correction has been made.

In this subsection, I do not see any specific studies related to G. orontii on Hydrangea. The cited work is referring to various biological control agents on other powdery mildew or pathogens (Botrytis) on other hosts. (Biological Control;L128) Botrytis (L133), Pea and vineyard (L134), In zinnia (L140)

Thank you for your comment. There is much lacking in the way of biological control studies in hydrangea specifically, so other studies were drawn upon that can be applied to hydrangea in future studies.

Same comment as in the above section – there is not much information on G. orontii control on Hydrangea except for one sentence. I suggest the authors consider providing more detailed information on the control of G. orontii in Hydrangea specifically through chemical means (Chemical control section; L144)

Thank you for the comment, additional information on powdery mildew of bigleaf hydrangea has been added to the chemical control section. (L194-197)

Please cite each statement in this section. Also provide information on how to manage G. orontii on Hydrangea by cultural means in various settings (land scape, greenhouse etc). (Cultural Control section; L168)

Thank you, the cultural control section has been expanded on and supplemented with more citations. Many cultural control references are general to host-pathogen interactions overall, with little being available that is very specific to G. orontii of bigleaf hydrangea. These statements reflect applications that can be used for future studies on G. orontii on bigleaf hydrangea. (L213)

This section also provides general information and deviates from the main topic of interest. (Section 5; L184)

Thank you for your comment. The information within section 5 is meant to be what has been found for molecular mechanisms of powdery mildew resistance – which there are not many studies over specifically G. orontii on bigleaf hydrangea. This section aims to discuss avenues of research that can be applied to bigleaf hydrangea.

Reduce the number of epidemics or reduce powdery mildew severity? Could you please be more specific? (L217)

Thank you, the term epidemics was used to due to speaking of multiple plants in a single setting  vs the severity of one singular plant. The statement was expanded on to specify that the statement is meant for a group of plants vs. an individual plant. (L280)

Same comment as above, not much information on the host and pathogen of interest are discussed here. (Section 6; L241)

Thank you for the comment. We greatly expanded section 6, removed some details regarding PM inoculation, and rearranged the text to focus on hydrangea breeding for PM resistance (lines 319-442).

The sentence you provided is incomplete. You can complete the sentence like this: “Powdery mildew fungi synchronize their reproductive and developmental phases with the life cycle of the host plant.” (L244)

Thank you for the suggestion, corrected as advised.

Since this is from roses, are there any studies on H. macrophylla? (L296 (ref37))

Thank you for the comment. As far as molecular studies of powdery mildew on bigleaf hydrangea go, we have found little to no literature related to the topic. The aim is to apply information from other horticultural crop studies and apply them to bigleaf hydrangea. Candidate genes, linked markers, and other outputs from transcriptional studies are listed in lines 390-394, but as yet have not been used in a completed hydrangea breeding improvement project.

Reviewer 5 Report

Comments and Suggestions for Authors

This manuscript has systematically reviewed the biology of Powdery mildew in hydrangea, powdery mildew in Hydrangea production, Molecular mechanisms of powdery mildew resistance, and Breeding for powdery mildew resistance in hydrangea. Some suggestions were made in order to improve the quality of this review. Therefore, a major revision is proposed.

General remarks

1. In “Abstract” part, the author should present the main point what the author will review in the manuscript.

2. L75-78, During the pathogen of Blumeria graminis infection process, the author can make a illustration to demonstrate the conidium germinating, cuticular peg, appressorium, Protrusions………production and formation, and how to infect the Protrusions. 

3. Fig.2, the author should provide which period to cause the different disease symptom, for example, how many days does Protrusions begin to displays a low amount 120 of infection after flowering, how many days does Protrusions displays a moderate amount of infection.

4. Line 134-135: Trichoderma is attributed to fungi, the author mention fungi again, is there any repetition. Please check it.

5. Line 138-139: Plant extracts are mainly attributed to chemical control?

6. Please check this sentence “this typically is by reducing/stopping spore production”.

7. Crispr/cas9 is one of the most important Genome editing methods, which is also the future innovative technology for breeding. The author should provide some related reference.

8. For “Conclusion” part, the author did not cover the main point the author review in this review.

 Specific remarks

1. L 212: change “We” to “we”.

2. L212: change “in helping us” to “to help us”.

3. L270: “in vitro” should be italic.

4.L319: Powdery mildew is not plant pathogen, please check it

5. L391: “make it significant” is not very correct, please change it.

6. reference 1, 7, 8, 9, please provide the doi No.

Comments on the Quality of English Language

Moderate editing of English language required

Author Response

Reviewer 5

This manuscript has systematically reviewed the biology of Powdery mildew in hydrangea, powdery mildew in Hydrangea production, Molecular mechanisms of powdery mildew resistance, and Breeding for powdery mildew resistance in hydrangea. Some suggestions were made in order to improve the quality of this review. Therefore, a major revision is proposed.

General remarks

  1. In “Abstract” part, the author should present the main point what the author will review in the manuscript.

Thank you for the comment, information added into the abstract to clarify the main point of the manuscript. (L21-23)

  1. L75-78, During the pathogen of Blumeria graminisinfection process, the author can make a illustration to demonstrate the conidium germinating, cuticular peg, appressorium, Protrusions………production and formation, and how to infect the Protrusions

Thank you for your comment. An illustration has been made to show the infection process of Blumeria graminis and has been added as a new Figure 1 and the rest of the figures have been shifted down in number, respectively.

  1. Fig.2, the author should provide which period to cause the different disease symptom, for example, how many days does Protrusionsbegin to displays a low amount 120 of infection after flowering, how many days does Protrusions displays a moderate amount of infection.

Thank you for the comment. A new statement including the time of first primary germ tube and disease development timeline have been added. (L96-100)

  1. Line 134-135: Trichodermais attributed to fungi, the author mention fungi again, is there any repetition. Please check it.

Thank you for your comment. The second mention of fungi has been removed.

  1. Line 138-139: Plant extracts are mainly attributed to chemical control?

Thank you for your comment. In the literature I’ve searched, plant extracts are a good replacement for chemical control, but are a biological control.

  1. Please check this sentence “this typically is by reducing/stopping spore production”.

Thank you for the comment. The sentence was updated from “In fungicides, this typically is by reducing/stopping spore production…” to “In fungicides, the MOAs typically work by reducing…” so that the sentence is complete. (L191)

  1. Crispr/cas9 is one of the most important Genome editing methods, which is also the future innovative technology for breeding. The author should provide some related reference.

Thank you. A new section “Engineering PM resistance in bigleaf hydrangea” was added to cover applications of, and barriers to, transgenic breeding and gene editing in hydrangea, as suggested (lines 418-442).

  1. For “Conclusion” part, the author did not cover the main point the author review in this review.

The conclusion has been expanded to include the main points of the review. (L444-456)

 Specific remarks

  1. L 212: change “We” to “we”.

Thank you, the change has been made.

  1. L212: change “in helping us” to “to help us”.
    Thank you for the comment, the correction has been made.
  2. L270: “in vitro” should be italic.
    Thank you, the correction has been made.

4.L319: Powdery mildew is not plant pathogen, please check it

Thank you for your comment, the term pathogen was switched to disease.

  1. L391: “make it significant” is not very correct, please change it.

Thank you for your comment, the statement was corrected.

  1. reference 1, 7, 8, 9, please provide the doi No.

Thank you for the comment, the DOIs were added to the references.

Round 2

Reviewer 3 Report

Comments and Suggestions for Authors

The new title is more appropriate and linked to the object of the manuscript.

It is obvious that the content has been greatly improved in the revised version of the paper. New data supplement the overview with information related to the topic. Rearrangements of sections, examples, all new references strictly connected to bigleaf hydrangea and powdery mildew, improvements into the breeding strategies section, expand the range of investigations included – all these changes make the article more useful for readers and suitable for the Special Issue. The new figures are helpful, informative and detailed. The revised version of Conclusions corresponds to the manuscript content.

There is still one small drawback that could be improved easily. Some words are repeated frequently in the text, occasionally even 2 or 3 times in one sentence. They could be replaced by synonyms. For example, in the section “Control of powdery mildew” the word “control” could be changed by ”manage” in some sentences to avoid numerous repetitions.

Some other suggestions are noted below. My advice to the authors is to go through the entire text once more.

Page 2, rows 46-48

The word “cause” is used 3 times in one sentence. Choose some synonyms to improve the sentence. For instance: “….leads to a decrease in yield, resulting in economic loss.” or similar.

Pages 2/3, rows 88-113

This part could be separate paragraph.

Page 6, rows 197-199

“There have been various studies conducted studying the control of powdery mildew of bigleaf hydrangea by fungicides with successful control” - the sentence need of revision to be more precise and clear (For instance: There have been various studies conducted evaluating the control of powdery mildew of bigleaf hydrangea by fungicides with successful results/or outcome, menagement)

Page 7, rows 226

“prevent infection” instead of “avoid infection”

Page 7, rows 240

Start the sentence with “They” instead of “These methods”.

Page 8, rows 315-316

“….in order to compare up and down regulated genes using RNA-seq and compared”

Page 8, rows 315-316

Use the term “marker assisted selection” instead of “marker aided selection”.

Page 9, rows 341-342

The sentence is not clear enough.

Page 9, rows 373-374

The sentence is not clear enough.

Should be Figure 4A instead of Fig 3A

Page 10, rows 381-383

“Interspecific hybridization provides an opportunity to understand which crosses and hybrids provide the best powdery mildew resistance and lead to novel plants that can be introduced that have natural powdery mildew resistance.”

Author Response

Reviewer 3

It is obvious that the content has been greatly improved in the revised version of the paper. New data supplement the overview with information related to the topic. Rearrangements of sections, examples, all new references strictly connected to bigleaf hydrangea and powdery mildew, improvements into the breeding strategies section, expand the range of investigations included – all these changes make the article more useful for readers and suitable for the Special Issue. The new figures are helpful, informative and detailed. The revised version of Conclusions corresponds to the manuscript content.

Thank you for your thorough reviews that led to a much improved manuscript. We know how long it takes to review manuscripts and truly appreciate your time.

There is still one small drawback that could be improved easily. Some words are repeated frequently in the text, occasionally even 2 or 3 times in one sentence. They could be replaced by synonyms. For example, in the section “Control of powdery mildew” the word “control” could be changed by ”manage” in some sentences to avoid numerous repetitions.

Changed as suggested throughout the manuscript e.g. lines 46-47, 54-56, 78-79, 178-179.

Some other suggestions are noted below. My advice to the authors is to go through the entire text once more.

We reviewed the text again to improve grammar, word use, and scientific language.

Page 2, rows 46-48: The word “cause” is used 3 times in one sentence. Choose some synonyms to improve the sentence. For instance: “….leads to a decrease in yield, resulting in economic loss.” or similar.

Changed as suggested to: “Approximately 70% of plant diseases are caused by fungi, which often leads to a decrease in yield and resultant economic loss.”

 Pages 2/3, rows 88-113: This part could be separate paragraph.

Changed as suggested.

 Page 6, rows 197-199: “There have been various studies conducted studying the control of powdery mildew of bigleaf hydrangea by fungicides with successful control” - the sentence need of revision to be more precise and clear (For instance: There have been various studies conducted evaluating the control of powdery mildew of bigleaf hydrangea by fungicides with successful results/or outcome, menagement)

Changed as suggested to say: “Various studies have shown successful control of powdery mildew on bigleaf hydrangea using fungicides.”

Page 7, rows 226: “prevent infection” instead of “avoid infection”

Changed as suggested.

Page 7, rows 240: Start the sentence with “They” instead of “These methods”.

Changed as suggested.

Page 8, rows 315-316:“….in order to compare up and down regulated genes using RNA-seq and compared”

Changed as suggested.

Page 8, rows 315-316: Use the term “marker assisted selection” instead of “marker aided selection”.

Changed as suggested.

 Page 9, rows 341-342: The sentence is not clear enough. Using a spore-suspension provides more consistent and even coverage but can result in poor germination, due to reduced viability of spores after the water-suspension process

Changed as suggested to: “Using a spore-suspension provides more consistent and even coverage of but can result in poor spore germination due to reduced viability of spores after the water-suspension process.”

Page 9, rows 373-374: The sentence is not clear enough. This can be observed through evaluation trials where plants are observed for disease severity over a set amount of time.

Thank you for your comment. This sentence was redundant and removed for clarity.

Should be Figure 4A instead of Fig 3A

Thank you, changed as suggested.

Page 10, rows 381-383: “Interspecific hybridization provides an opportunity to understand which crosses and hybrids provide the best powdery mildew resistance and lead to novel plants that can be introduced that have natural powdery mildew resistance.”

Changed to: “Interspecific hybridization provides an opportunity to understand which crosses and hybrids provide the best powdery mildew resistance and to develop new, resistant cultivars.”

Reviewer 4 Report

Comments and Suggestions for Authors

My comments can be found in the attached MS.

Comments on the Quality of English Language

My comments on the language can also be found in the attached MS.

Author Response

Reviewer 4

You ended this sentence with a fruit crop (L 87)

This has been corrected to crape myrtle instead of apple. (L 89)

Please check spelling here (L 113)

Thank you, spelling corrected.

Given the distinct life cycle of G. orontii and its interaction with hydrangea, the current figure might lead to potential confusion for readers seeking specific insights into this particular pathogen. Could you kindly consider revising or providing additional information that aligns more closely with the unique characteristics of G. orontii on hydrangea (L116)

Reviewer 5 requested that the B. graminis life cycle be described in an illustration since it is a commonly studied powdery mildew. However, the text has been clarified to mention that this is a powdery mildew of grasses and not of hydrangea.  (L 83-84 )

Italics please. (L 116)

Corrected.

This figure is also misleading to the reader. Although you mentioned that chasmothecia are not formed (lines 136-137), the figure depicts a life cycle that includes chasmothecia. (L119)

Thank you for your comment. The phrasing was changed to indicate that the asexual stage is the more typical stage and the sexual stage with chasmothecia is rarer but can occur. The figure text has also been updated to specify this to avoid any confusion by readers. (L120-121, 146-148)

If chasmothecia are not formed, please mention the possible reasons for their absence. (L 130)

Chasmothecia are rarely formed, which we corrected in the text to avoid confusion. (L120-121, 146-148)

I am curious about the reasons behind the name change from Erysiphe polygoni to Golovinomyces orontii.  Could you provide some insights into whether this change was prompted by a misidentification in earlier studies or if it resulted from additional taxonomic research. (L 134)

Thank you for the comment. The name changed was prompted by the taxonomy of powdery mildews changing and the way that these fungi are identified by characteristics. (L137-143).

Microscopic images could greatly enhance the visual understanding of the pathogen for readers.
Would it be possible to include micrographs of the powdery mildew, particularly showcasing its morphological features? This addition would contribute significantly to the overall clarity and completeness of the manuscript. (L 141)

While we do not currently have micrographs of powdery mildew, we have greatly expanded the description and cited a paper that does have micrographs and different time points after inoculation to provide an example of microscopic images. (L152-159)

While the figures of hydrangea plants with varying levels of powdery mildew infection provide valuable visual representation, it seems that their relevance for readers with a scientific background might be limited. I prefer to see the micrographs of the pathogen. (L 157)

Thank you, the text was greatly expanded on to describe these and paper referenced that has micrographs.  (L152-159)

I do not agree with this statement; it is more effective to implement preventative measures based on predictive disease models and apply fungicides proactively before symptoms appear. Taking early action can help minimize the impact of the disease and improve overall control. Since the symptoms/signs of powdery mildew appear after an infection is initiated 7-10 days before the visual appearance of the disease. (L 208)

Thank you for the comment. You are correct, it is more effective to implement preventative measures. I have changed the wording to reflect this, but also added in the downside to applying preventatively, such as the economic impact to growers and environmental impact. (L224-228)

Please add a couple of lines for clarification on how it helps in the avoidance of infection. (L 226)

Thank you for the comment, lines have been added for clarification that describe nitrogen helping the growth rate of the host plant and that it stimulates new growth that is more susceptible to powdery mildew infection. (L247-249)

May I suggest a more formal phrasing for a review context, such as, “This material should be eradicated to ensure the effective termination of the fungus.” (L 234)

Thank you for your comment, the sentence was changed as suggested.

In the case of powdery mildew fungi, which are known to be host-specific, the practice of separating crops may not necessarily contribute to the delay or reduction of epidemics. As these fungi exhibit host specificity, the spread is typically limited to plants within the same species. Therefore, the effectiveness of this strategy may vary based on the pathogen’s characteristics.  (L 282-284)

Thank you for the comment, the sentence was removed.

Reviewer 5 Report

Comments and Suggestions for Authors

The authors have modified all the comments, and can be accept after minor editing of English language. 

Comments on the Quality of English Language

Minor editing of English language is required. 

Author Response

Reviewer 5

The authors have modified all the comments, and can be accept after minor editing of English language.

Thank you for your thorough reviews that led to a much improved manuscript. We reviewed the text again to improve grammar, word use, and scientific language.

Minor editing of English language is required.

Thank you for your comment. We reviewed the text again to improve grammar, word use, and scientific language, e.g. lines 46-47, 54-56.

Round 3

Reviewer 4 Report

Comments and Suggestions for Authors

My comments can be found in the attached MS.

Comments on the Quality of English Language

My comments can be found in the attached MS.

Author Response

Thank you for your thorough review and comments. Within the manuscript, edits are highlighted in blue for extra clarity; reference numbers that have been changed are highlighted in yellow.

The current phrasing suggests that most powdery mildews cannot be cultured, which might be misleading. It could be helpful to clarify that while culturing is generally challenging due to their biotrophic obligate nature, if there are exceptions or instances where some powdery mildews have been successfully cultured on an artificial media, please provide them, as I am not aware of any. This clarification would contribute to a more precise and accurate representation of the current understanding in the field. (L79).

Thank you for your comment. I have added a statement clarifying that Blumeria graminis has successfully been cultured. (L 80-81)

Could you please rewrite this as the statement in the current form is misleading as you ended with “changing”? “The name change occurred because there were extensive changes in the taxonomy of powdery mildew fungi.” (L137-138)   

I appreciate the comment, thank you for the advice. The statement was changed as advised. (L140-141)

It’s (L 152)

Corrected, thank you. (L156).

Please consider this revision “Three days after inoculation, multiple hyphae emerge from a conidium, forming a branched hyphal structure that will subsequently develop into a lesion” (L157-158)

The change has been made as advised. (L161-162)

“colony” typically implies a collection of individual organisms or structures. In the context of powdery mildew, it is more common to describe the visible growth or accumulation as a “lesion” or a “spot” rather than a “colony.” I would like to recommend that the authors consider consulting recent literature or seeking expert guidance to ensure the accurate and precise use of terminology in the field. This step can significantly enhance the clarity and credibility of the manuscript. (L157-159)

Thank you for the comment, colony was replaced by lesion or spot (L 162, 196, 365)

Please consider this revised version “Biological control methods stand out for their effectiveness in minimizing adverse environmental impacts.”

The revised version has been implemented. (L186-187)

The term “host organisms” could be revised to simply “hosts” for increase clarity and conciseness. (L189)

Deleted ‘organisms,’ thank you for the comment. (L192)

This is a vague ending, please consider revising the statement. Suggested revision: “Biocontrol of powdery mildew was achieved on various hosts, including Trichoderma isolates, yeasts, mycophagous arthropods, mycolytic bacteria, and additional effective biological agents.” (L190)

The revised statement has been implemented, thank you. (L192-194)

I have a concern about the phrase “disease must be observed”, Here’s a revised version: “For realistic and cost-effective applications, it is essential to monitor the disease actively rather than relying solely on preventative measures.” (L222)

Thank you for your comment, the statement has been revised as advised. (L225-227)

While preventive fungicide applications are indeed effective in controlling powdery mildew symptoms, it’s essential to acknowledge that achieving complete prevention might be a challenging goal. (L225)

Thank you for the comment, I have revised the statements to point out that control is a challenging goal to meet despite the best methods being those that are preventative. (L228-229)

To enhance clarity, you might consider rephrasing the sentence:

"Nonetheless, certain non-fungicidal products, such as chitosan, have been employed. These products possess promising commercial value as they offer broad-spectrum plant protection in an environmentally friendly manner. (L232-234)

The sentence has been rephrased as advised, thank you (L 236-238).

You may consider the revised version “Selecting cultivars that exhibit resistance to the specific pathogen, if available, helps mitigate the spread of pathogens to susceptible plants.” (L303).

Thank you for the comment, the statement has been adjusted as advised. (L307-308)

Round 4

Reviewer 4 Report

Comments and Suggestions for Authors

My comments can be seen in the attached MS.

Comments on the Quality of English Language

My comments can be seen in the attached MS.